# Molecular characterization of the effects of heat shock on the infection cycle progression and productivity of the baculovirus expression vector system

Enrique Paz-Cortés[1], Ana Ruth Pastor[1], Roberta Salinas-Marín[2,3], Octavio T. Ramírez[1,2], Laura A. Palomares[1]*

**1** Departamento de Medicina Molecular y Bioprocesos, Instituto de Biotecnología, Universidad Nacional Autónoma de México, Cuernavaca, Morelos, Mexico, **2** Laboratorio Nacional para la Producción y Análisis de Moléculas y Medicamentos Biotecnológicos, Instituto de Biotecnología, Universidad Nacional Autónoma de México, Cuernavaca, Morelos, Mexico, **3** Laboratorio de Glicobiología y Diagnóstico Molecular, Centro de Investigación en Dinámica Celular, Universidad Autónoma del Estado de Morelos, Cuernavaca, Morelos, Mexico

* laura.palomares@ibt.unam.mx

## Abstract

Baculoviruses are widely utilized in biotechnology for various purposes, including recombinant protein expression, antigen presentation, vaccine production, as biopesticides, and as gene therapy vectors. The productivity of the baculovirus expression vector system (BEVS) is significantly affected by the condition of the host cell. However, the impact of host cell stress on the complex baculovirus infection cycle remains not fully understood. This study examines the effects of three gradual heat shock treatments on the production of recombinant protein and viral titers in Sf9 cells (*Spodoptera frugiperda*) infected with a recombinant baculovirus AcMNPV with fluorescent reporters under late (*vp39*) and very late (*polh*) promoters. The heat shock regimens applied before infection were 30°C for 2.5 hours, 37°C for 2.5 hours, and constant 30°C, combined with prostaglandin A1 (PGA1) to enhance the cellular stress response. Significant differences in viral progeny and baculovirus genome replication were observed. Notably, a constant 30°C heat shock increased early viral titers but decreased late-stage yields. Using flow cytometry, we monitored the signal from the two fluorescent reporters and found that some heat shock conditions differentially accelerated or increased their timing or expression levels, with different patterns for each reporter. Additionally we identified, cloned, and sequenced two inducible HSP70 genes from *S. frugiperda* to track their expression throughout infection, providing insights into the cell's stress response and the effect of PGA1. These findings suggest that modulating the host heat-shock response can improve baculovirus production and offer insights into the host-virus relationship for new elements or strategies to improve BEVS productivity.

**Data availability statement:** GenBank under accession numbers PQ108612 and PQ108613. All other relevant data are within the manuscript and its Supporting Information Files.

**Funding:** Universidad Nacional Autónoma de México DGAPA-UNAM PAPIIT IT200824 Enrique Paz Cortés received a CONAHCYT fellowship 270335. The funders had no role in study design, data collection and analysis, decision to publish, or preparation of the manuscript.

**Competing interests:** The authors have declared that no competing interests exist.

## Introduction

Baculoviruses are double-stranded DNA viruses that infect arthropods of the order Lepidoptera. Among these, *Autographa californica* multiple nucleopolyhedrovirus (AcMNPV) is one of the most extensively used, with biotechnological applications ranging from agricultural pest control to the expression of complex eukaryotic recombinant proteins, production of viral vectors, vaccines, and gene delivery in gene therapy [1,2]. The insect cell-baculovirus expression vector system (BEVS) based on AcMNPV has led to commercially successful products. However, further improvements in BEVS require a deeper understanding of the host-virus relationship and how variations in host and viral factors affect the system performance. This study investigates how stress stimuli induce beneficial host cell alterations that enhance BEVS productivity of recombinant protein or baculoviral progeny.

The AcMNPV infection cycle is characterized by four overlapping phases: immediate early, early, late, and very late. This is based on key events in baculovirus replication cycle and the expression timing of phase-specific genes (Fig 1) [2]. Throughout this cycle, two distinct types of viral progeny are produced. The first type, budded viruses (BV), are infectious to other cells and begin budding from the membranes of infected cells around 15 hours post infection (hpi), during the late phase (around 6 to 18 hpi). BV production continues into the very-late phase, although at reduced levels. The very late phase starts approximately at 18–24 hpi and is characterized by a marked increase in the expression of very-late-phase genes, which continues until cell lysis [3,4]. During this last phase, the second type of viral progeny, occluded viruses (OCVs), are formed. OCVs are nucleocapsids embedded in a polyhedrin matrix, which protects them from environmental factors and are infectious when released in the insect gut upon ingestion [2,5]. In the BEVS, the *polh* gene, which codes for polyhedrin, is usually replaced with a gene of interest (GOI) to harness the high activity of its promoter during the infection very-late phase [6]. Thus, the main products of the BEVS during the baculovirus cycle are BV, produced primarily during the late phase, and recombinant protein, predominantly expressed during the very-late phase when the GOI is driven by the *polh* promoter.

In addition to using the BEVS for recombinant protein expression, BV have applications, such as protein or antigen display or as versatile gene delivery vectors capable of entering cells from multiple species, including mammalian cells, making them useful for gene therapy [7–9]. It has been shown that there is an interplay between the production of BV and the expression of the GOI under the *polh* promoter, and that stress can alter the balance between the late and very-late phases of the infection cycle [10,11]. The virus infection cycle progression through its four phases is orchestrated by several promoters and regulated by host and viral transcription factors [12], but the impact of other cellular elements or conditions is still not fully understood.

Different cell lines from various organisms are utilized with AcMNPV in the BEVS, showing varied performance among them. Some of the most used are *Spodoptera frugiperda* (Sf)-derived cell lines, such as Sf9 and Sf21, and *Trichoplusia ni*-derived cell lines, such as High Five and Tnao38. Sf cell lines are recognized for their ability to support high-titer baculovirus replication, making them a preferred choice for viral stock production. Conversely, *T. ni*-derived cell lines typically yield higher levels of recombinant proteins when using the *polh* promoter, emphasizing their suitability for protein production [13,14]. Sf9 cells were selected for this study due to their robust viral replication capacity, making them appropriate for studying host factors or conditions, such as stress or the over expression of Sf HSP70 genes, during early stages of the infection cycle.

Studies in Sf cells have demonstrated that cellular heat shock response genes, particularly those in the HSP70 family, are significantly upregulated during AcMNPV infection, in contrast to the general downregulation of most other cellular genes [15–18]. Hsc/Hsp70 proteins

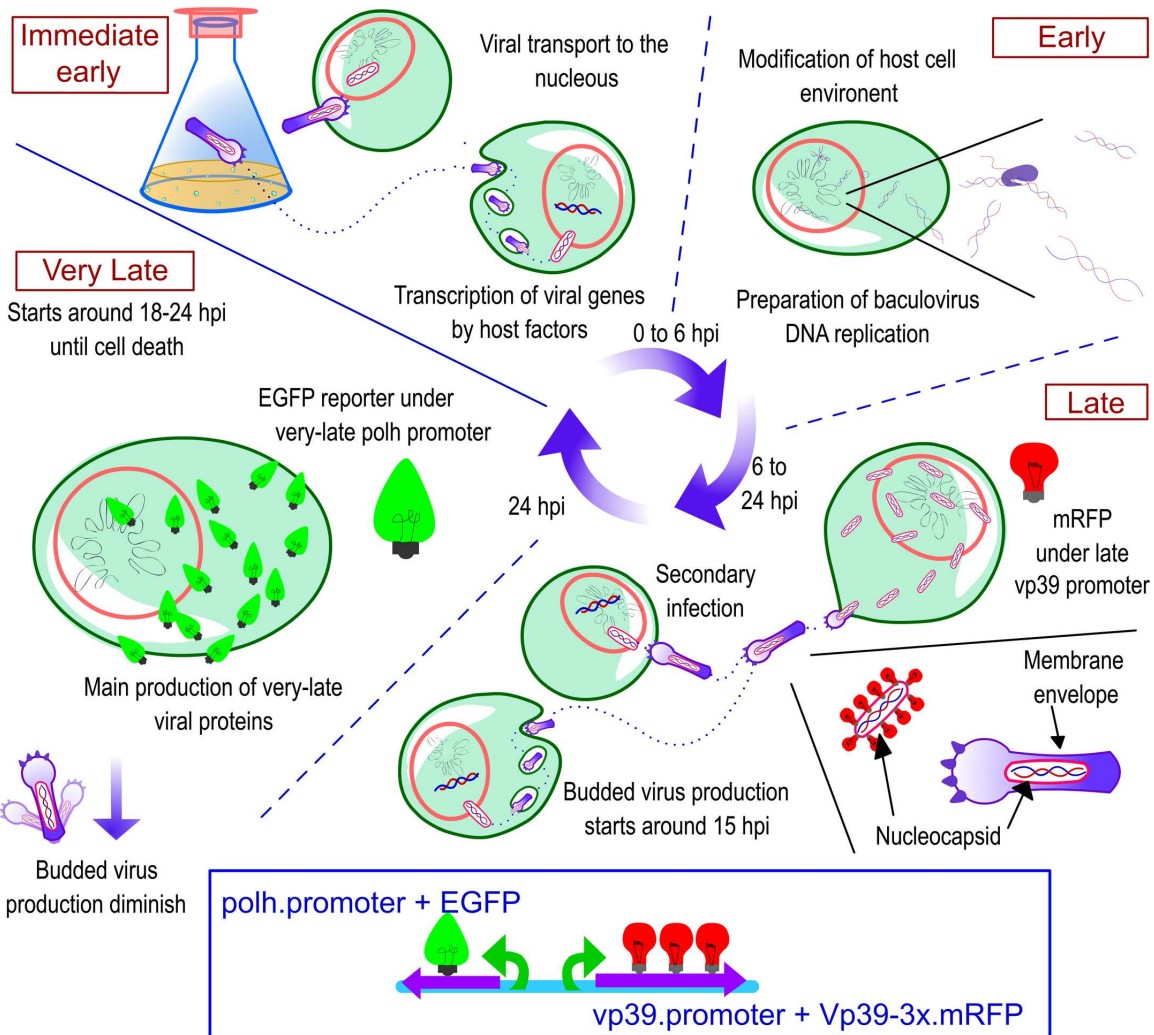

**Fig 1. The four phases of AcMNPV infection cycle and the design of dual reporter baculovirus (DRBac).** We used the DRBac in this study to monitor fluorescent protein reporters under late and very late phase promoters. After budded viruses (BV) are added to a cell culture for primary infection, capsids are transported to the nucleus, initiating the immediate-early phase with baculovirus regulatory genes transcribed by host factors. The early phase follows with the production of viral factors and polymerases that prepare for baculovirus DNA transcription and replication. The late phase begins with genome replication and BV production, spreading the infection to other cells in the culture. In the DRBac, the activity under a late-phase promoter (*vp39*) is monitored by the expression of the monomeric red fluorescent protein (mRFP). BV production declines during the very-late phase, and transcription of very-late genes, such as *polh* and *p10*, is significantly upregulated starting about 18-24 hours post infection (hpi). The activity of the *polh* promoter is tracked through the expression of the green fluorescent protein (EGFP). This promoter is commonly used for recombinant protein production due to its high activity.

are associated with early viral life cycle events, including genome replication, transcription, and viral proteins stability or transport [15–20]. Inhibition of Hsp70 gene transcription or protein activity significantly reduces viral DNA synthesis and BV production, with the magnitude of this reduction depending on the timing of inhibitor application, earlier inhibition results in a greater decrease in viral replication [15,19,20]. Similar results have been reported in *Bombix mori* infected by its *B. mori* nucleopolyhedrovirus [21].

The induction of *S. frugiperda hsc/hsp70* genes by baculovirus infection is documented at both mRNA and protein levels [17,18,22–28]. A synergistic effect of infection and thermal

stress on the induction of Hsp70 proteins has been observed [15], although their effects on the progression of the infection cycle and viral productivity remain scarcely investigated. The sequence of some *hsc/hsp70* genes of *S. frugiperda* have been reported but are still not fully identified or characterized. A better understanding of the role of cellular factors that participate in and regulate the infection cycle may be useful for expanding our knowledge of the AcMNPV-host cell relationship and enhancing their biotechnological applications.

This study aims to understand the effect of stress in the production and interplay between the late and very-late phase products of the AcMNPV infection cycle, using two fluorescent reporter genes, each under a different promoter, one that starts expression on the late phase and the other on the very-late phase of infection (Fig 1). We investigated the impact on productivity and infection cycle dynamics of distinct host stress-inducing conditions, using three heat shock treatments before infection. The cellular heat-shock stress response was monitored by following the expression of two *hsp70* genes from *S. frugiperda,* partially identified as over-expressed during baculovirus infection [18]. Each condition generated distinct *hsp70* genes expression profiles, all generally peaking in expression at 12 hpi, and the highest values under a condition of constant heat shock. The heat shock response of *S. frugiperda* cells was analyzed in the context of baculovirus infection and heat shock, focusing on the progression of the viral cycle and the kinetics of viral main products. These insights may be important for improvements in the BEVS as well as for other applications of budded viruses, such as gene therapy.

## Materials and methods

### Cell line and recombinant baculovirus

Sf9 insect cells were cultured in 125 mL shake flasks in 30 mL of PSFM medium (WAKO, Japan) at 27°C and 110 rpm. Cell concentration was determined using a Coulter Counter (Beckman Coulter, USA). Viability was assessed by trypan blue exclusion in a Neubauer chamber. Exponentially growing cultures with viability above 95% were used in all experiments. The recombinant dual reporter baculovirus AcMNPV (DRBac) used in this study was previously described in [10]. It was designed to monitor baculovirus cycle progression and activity under the late and very-late promoters of *vp39* and *polh* genes (Fig 1). It contains three copies of the monomeric red fluorescence protein (mRFP) fused to a copy of the *vp39* baculovirus gene and its late-phase promoter. Additionally, it contains the enhanced green fluorescent protein (EGFP) gene under the very-late *polh* promoter (Fig 1).

Baculovirus stocks were passage two and were used within two months of titration. Viral titers were determined with a 50% tissue culture infectious dose (TCID50) assay via the Reed-Muench Method [29], using fluorescence as a positive infection marker in 96-well plates seeded with Sf9 cells. The conditions of seeded cells and calculations to transform TCID50 values to plaque forming units (PFU) were like those described in [30], using the formula derived from the Poisson distribution PFU/mL = TCID50/mL × 0.69. Fluorescent cells were detected after five days of infection using a TE300 inverted microscope (Nikon Co.).

### Infection assays, heat shock treatment and PGA-1 addition

Eight different infection conditions were evaluated in triplicate and are described in Table 1. The control temperature was 27°C, and three different heat shock stimuli: 30°C or 37°C for 2.5 hours or constant 30°C were administered to Sf9 cells three hours before infection with the DRBac. For heat shock, triplicate cultures at each condition were transferred from an incubator at 27°C to incubators at 30°C or 37°C. After 2.5 hours, culture flasks were returned to 27°C for 30 minutes before infection. Cells were infected at a concentration of 0.8 x 10$^6$ cells mL$^{-1}$, using a multiplicity of infection (MOI) of 1 PFU cell$^{-1}$. To further induce Hsp70 genes

**Table 1. The eight growing conditions generated by combining different heat shock stimuli and PGA1 addition.**

| | | Heat shock | | |
|---|---|---|---|---|
| | Condition | Temperature | Duration | PGA1 |
| 1 | 27°C | Control 27°C | – | – |
| 2 | 27°C + PGA1 | Control 27°C | – | +PGA1 |
| 3 | Hs_30°C | 30°C | 2.5 h | – |
| 4 | Hs_30°C + PGA1 | 30°C | 2.5 h | +PGA1 |
| 5 | Hs_constant_30°C | 30°C | 99 h | – |
| 6 | Hs_constant_30°C + PGA1 | 30°C | 99 h | +PGA1 |
| 7 | Hs_37°C | 37°C | 2.5 h | – |
| 8 | Hs_37°C + PGA1 | 37°C | 2.5 h | +PGA1 |

Heat shock was applied at 3 hours before infection while cultures at conditions 5 and 6 were exposed to 30°C also after infection for the indicated time.

transcription, 4 μg mL$^{-1}$ of prostaglandin A1 (PGA1, Cayman Chemical, CAS 14152-28-4) diluted with DMSO was added 6h before heat shock for an additional conditions set. DMSO concentration was less than 0.01% of the total culture volume.

## Baculovirus genome quantification by real-time quantitative PCR (qPCR)

Budded virus genomes (BV-genomes) in culture supernatants were quantified by real-time qPCR using primers specific to the EGFP gene, present both in the DRBac and in a reference plasmid used to generate a standard curve. Primers sequences and their efficiency are detailed in Table 2 in S1 File. Baculovirus DNA was extracted from 100 μL of supernatant using the PrepSEQ™ Nucleic Acid Extraction Kit (Applied Biosystems). The final extraction volume was 100 μL, which was subsequently diluted 1:20 in water to provide 1 μL per qPCR reaction. The assay utilized SYBR™ Green PCR Master Mix and was conducted in a QuantStudio™ 5 Real-Time PCR System (Applied Biosystems). EGFP gene copy numbers in each supernatant sample were determined using the QuantStudio™ Design and Analysis Desktop Software.

## Heat shock gene identification, cloning, and sequencing

The selection of *hsp70* genes in *S. frugiperda* cells was based on Expressed Sequence Tags (EST) previously associated with *hsp70* genes reported as highly upregulated host genes during AcMNPV infection [18]. EST SFL00008-5-1 and Sf2H01963-5-1 were retrieved from SPODOBASE [31], and compared to transcriptomic data in GenBank using BLAST. We designed primers to amplify the coding sequences of two genes based on nucleotide records with the highest similarity to these ESTs and used complementary DNA (cDNA) as template. The primers, hsp70-FWD and hsp70-REV 1 and 2, are listed in Table 1 in S1 File.

Total RNA was extracted from Sf9 cells six hours after a 2.5-hour heat shock at 37°C using the Quick-RNA Microprep kit (Zymo), integrity assessed by 1% agarose gel electrophoresis, and quantified with a NanoDrop™ 1000 (Thermo Scientific). cDNA was synthesized using a polyT primer following the RevertAid First Strand cDNA Synthesis Kit (Thermo Scientific). A 5 μL cDNA aliquot was used for PCR with hsp70-FWD and hsp70-REV 1 and 2 primers. The resulting ~2 Kb PCR products were gel-purified and cloned into the pJet vector using the CloneJet™ PCR Cloning Kit (Thermo Scientific). Initial sequencing was performed using the original primers with Sanger sequencing at the DNA Synthesis and Sequencing Unit at the Institute of Biotechnology (UNAM, Mexico). Based on the results internal primers were designed for complete gene coverage sequencing and RT-qPCR analysis. Sequences

were assembled into a consensus CDS using Gap5 software [32]. The genes were identified as *hsp70Ab* and *hsp68*, based on their best BLAST matches to *Drosophila melanogaster* sequences, and were annotated in GenBank with accession numbers PQ108612 and PQ108613, respectively.

## Relative quantification of heat shock gene transcripts by reverse transcription real-time qPCR (RT-qPCR)

Primers for RT-qPCR were designed using the BLAST-primer tool [33], with RefSeq mRNA data from *S. frugiperda* for specificity analysis. Primers targeted the annotated *hsp70Ab* and *hsp68* CDS, with the 28s ribosomal RNA gene (XR_004783999.1, GenBank) as internal reference. Due to multiple genes and isoforms with sequences similar to Sf2H01963-5-1 EST and the *hsp70Ab* CDS in the BLAST search, primers were designed for a common region, referred to as the *hsp70Ab*-group. Details of the BLAST search results, primer sequences, efficiency, and targets according to BLAST-primer are provided in S1 File.

For total RNA extraction, 350 µL of cultures were collected and immediately mixed with 350 µL of 2x DNA/RNA Shield™ (Zymo Research) at the indicated time points, and frozen at -80°C until extraction using the Quick-RNA™ Microprep kit (Zymo Research). RT-qPCR amplification was performed with primers at a final concentration of 150nM in a 20 µL reaction containing 5ng of RNA and 10 µL of Luna® Universal One-Step RT-qPCR (New England Biolabs) in the QuantStudio™ 5 Real-Time PCR System (Applied Biosystems) at 55°C for 10 min, 95 °C for 60s, followed by 40 cycles at 95°C for 10s, and 60°C for 60s. Relative gene expression was calculated with the $2^{(-\Delta\Delta Ct)}$ method using the 28s gene as the internal reference. Uninfected Sf9 cells cultured for 72 hours were the reference for normalization across different experiments and microplates.

## Quantitative fluorescence analysis by flow cytometry

Cells were harvested from 500 µL of culture by centrifugation and subsequently fixed in 1% paraformaldehyde in phosphate-buffered saline (PBS) overnight at 4°C. Prior to analysis, cells were washed and resuspended in cold PBS. Cells were analyzed in a BD Accuri™ C6 Plus flow cytometer (BD Accuri Cytometers, San Jose, CA, United States), capturing 10,000 events per sample. Fluorescence intensities of EGFP and mRFP in cells were collected using the fluorescein isothiocyanate (FITC) and phycoerythrin (PE) emission channels, respectively. Three biological replicates were analyzed for each condition. Data was processed using the FlowJo™ software (BD Life Sciences). Gating and compensation were performed using control cells infected with baculovirus expressing either EGFP or mRFP alone to set fluorescence positivity thresholds and to minimize signal overlap (more information is provided in Fig 1 in S3 File). Cells were categorized into four groups based on their fluorescence profiles: non-fluorescent cells (mRFP−/EGFP−), cells with only EGFP signal (mRFP−/EGFP+), cells with only mRFP signal (mRFP+/EGFP−), and cells with signals for both mRFP and EGFP (mRFP+/EGFP+). The geometric mean of fluorescence intensity in cells for mRFP and EGFP was used to compare groups and conditions.

## Statistical analysis

Statistical significance was set at $p < 0.05$. Results are presented as mean or geometric mean ± standard deviation (SD). All experiments were performed with three independent biological replicates. Statistical analysis was conducted using the GraphPad Prism software. A two-way ANOVA followed by a Tukey's multiple comparisons test was used to assess differences between the various treatments. Pearson's correlation coefficient was used to evaluate the

correlation between variables, with significance determined by the corresponding t-value and p-values.

## Results

### Heat shock design and administration on cell cultures

We designed three heat shock stimuli to evaluate the impact of various stress conditions on baculovirus infection dynamics and productivity of Sf9 cells, which are typically grown at 27°C (control temperature). The conditions were: moderate heat shock at 30°C for 2.5 hours (Hs_30°C), high heat shock at 37°C for 2.5 hours (Hs_37°C), and continuous culture at 30°C (Hs_constant_30°C) (Table 1). A constant 37°C condition was not used, as cells did not duplicate at this temperature. The intended gradual temperature increase was obtained by transferring culture flasks from 27°C to an incubator at either 30°C or 37°C. It was determined that the target temperature was reached approximately 30 and 60 minutes after transfer, respectively. Additionally, prostaglandin A1 (PGA1), known to induce the transcription of genes regulated by the heat shock factor protein [20], was added to the four conditions at a final concentration of 4 µg mL$^{-1}$ to further induce the transcription of *hsp70* genes (Table 1). After the heat shock, cultures were allowed to return to 27°C before adding the baculovirus.

We used a multiplicity of infection (MOI) of 1 plaque-forming unit per cell (1 PFU cell$^{-1}$) because it allows for secondary infection, which was important for our objective of identifying additional treatment effects on infection dynamics. At an MOI of 1, we expect approximately half of the cell population to be infected at the onset of infection [6], ensuring representative results while still permitting secondary infections. None of these heat shock stimuli or PGA1 addition altered the normal growth or viability of uninfected cells (Fig 1A, 1B in S2 File). Only at 72 hpi, we observed that PGA1 addition increased cell viability of infected cells by $12 \pm 5\%$, compared to their counterpart conditions without PGA1 (Fig 1D in S2 File).

### Heat shock stimuli effects on baculovirus titers and genome copy numbers

Two different measurements of viral progeny production were obtained from all conditions at 24, 48 and 72 hpi, viral titers and budded virus genome copy numbers (BV-genomes), to determine if there were differences between viral DNA genome replication and viral titers. The resulting titers and BV-genomes obtained under the eight conditions are shown in Fig 2A–2C, respectively.

The most significant differences were observed at 24 hpi (Fig 2A), caused by the heat shock stimuli rather than by the presence of PGA1. Less significant differences were found at 72 hpi, and none at 48 hpi (Fig 2B, 2C). The highest BV-genome copy numbers and BV titers were found at 72 hpi in practically all conditions. We measured PFU at 96 hpi, and only two conditions showed an increment in titers in respect to 72 hpi (Fig 2 in S2 File). BV-genomes were not measured at 96 hpi since cell viability in all conditions was too low, about 20% (Fig 1D in S2 File), which may lead to unreliable qPCR measurements. Interestingly, the conditions that resulted in the highest production of BV-genomes, at any time point, were different from those producing more PFU. In general, PFU titers did not change proportionally with genomes production.

Table 2 complements volumetric yield in Fig 2 with specific yields for PFU and BV-genomes production per cell, and the ratio among these two products for each sampled time and condition. At 24 hpi, the Hs_constant_30°C condition produced the highest titer, which was $3.83 \pm 1.96$ times higher than the control. In contrast, the highest BV-genome copy number was observed at the Hs_37°C condition, $8.19 \pm 2.37$ times higher than the BV-genome copy numbers of the control, and with the highest ratio of $118 \pm 72$ BV-genomes per PFU. Interestingly,

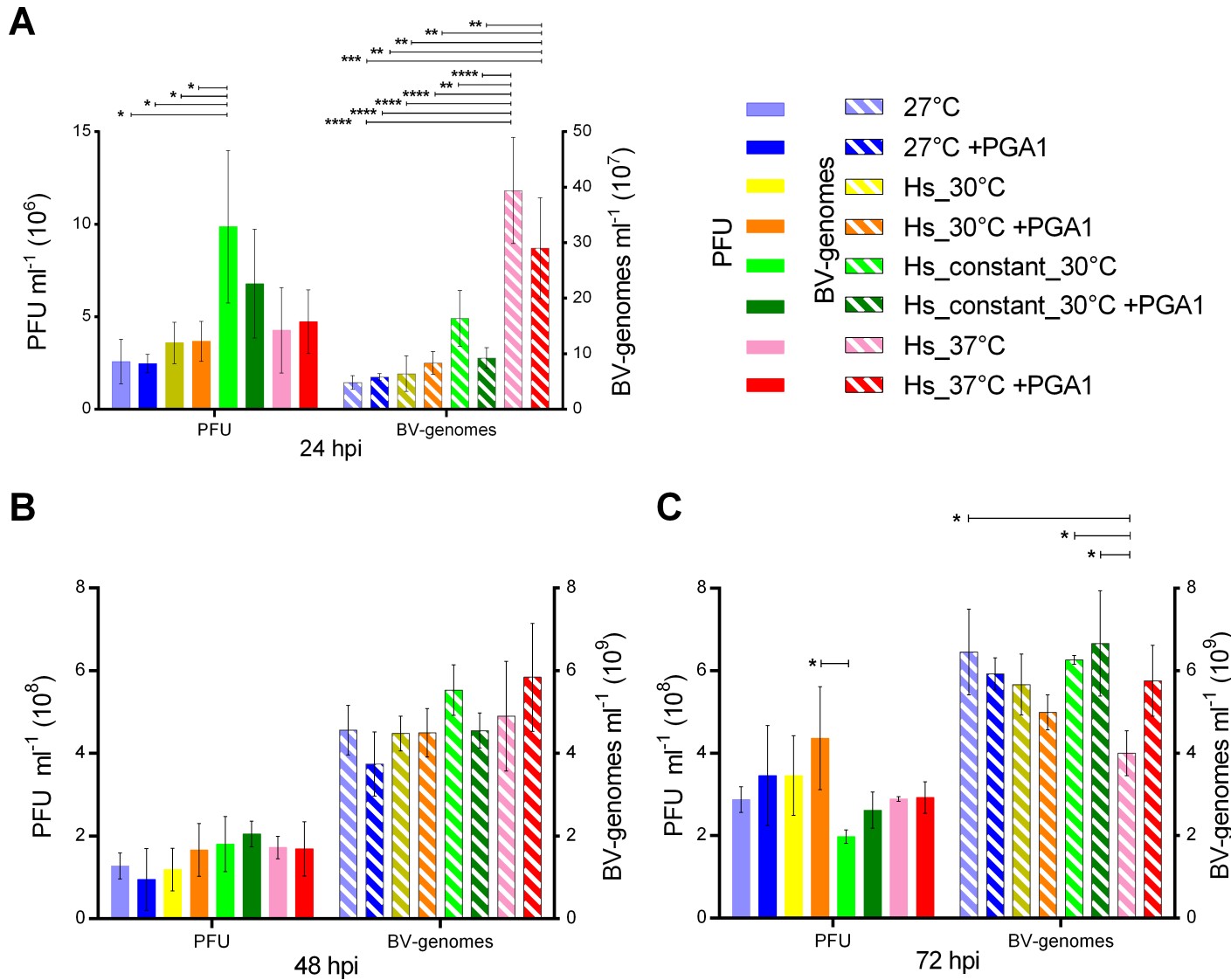

**Fig 2. Heat shock induced alterations in the production of viral progeny and baculovirus genomes.** Representation of volumetric yields for plaque forming units (PFU $ml^{-1}$) and baculovirus genomes (BV-genomes $ml^{-1}$) at different hours post-infection (hpi). (A): 24 hpi, (B): 48 hpi, (C): 72 hpi. Each bar represents an average of three biological replicates with standard deviations indicated by error bars. Statistical significance, determined by two-factor ANOVA and Tukey's multiple comparisons, is indicated by horizontal bars and asterisks (p: * ≤ 0.05, ** ≤ 0.01, *** ≤ 0.001, **** ≤ 0.0001).

the conditions with the highest concentrations of either PFU or BV-genomes at 24 hpi did not remain as the highest at later times. For conditions under Constant_30°C, the production of PFU was the highest at 24 hpi with the lowest ratio of BV-genomes per PFU, while titers at 72 hpi were the lowest of all conditions (Fig 2). All the other conditions reached similar titers at 72 hpi, and only Hs_37°C showed significantly lower levels of BV-genomes at this time.

## Flow cytometry evaluation to monitor infection progression

The progression of infection was followed by the fluorescence of infected cells, with monomeric red fluorescent protein (mRFP) as indicative of the late-phase promoter *vp39*, and green fluorescent protein (EGFP) indicating the activity under the very late *polh* promoter (Fig 1).

**Table 2. Specific yields and the BV-genomes copy number-to-PFU ratio (BV-genomes PFU$^{-1}$) at different times post-infection.**

| | PFU cell$^{-1}$ | | | BV-genomes cell$^{-1}$ | | | BV-genomes PFU$^{-1}$ | | |
|---|---|---|---|---|---|---|---|---|---|
| Time after infection, h | 24 | 48 | 72 | 24 | 48 | 72 | 24 | 48 | 72 |
| 27°C | 2 ± 1 | 59 ± 12 | 127 ± 3 | 36 ± 6 | 2093 ± 151 | 2921 ± 810 | 21 ± 11 | 37 ± 10 | 23 ± 6 |
| 27°C + PGA1 | 2 ± 0 | 43 ± 34 | 154 ± 59 | 44 ± 5 | 1697 ± 333 | 2618 ± 259 | 24 ± 7 | 52 ± 25 | 18 ± 5 |
| Hs_30°C | 3 ± 1 | 58 ± 24 | 160 ± 43 | 48 ± 23 | 2181 ± 129 | 2610 ± 216 | 20 ± 13 | 41 ± 12 | 17 ± 3 |
| Hs_30°C + PGA1 | 3 ± 1 | 90 ± 23 | 216 ± 80 | 66 ± 21 | 2501 ± 381 | 2447 ± 457 | 24 ± 10 | 29 ± 7 | 12 ± 5 |
| Hs_constant_30°C | 7 ± 2 | 82 ± 22 | 91 ± 8 | 112 ± 38 | 2622 ± 802 | 2898 ± 160 | 20 ± 14 | 34 ± 13 | 32 ± 3 |
| Hs_constant_30°C + PGA1 | 5 ± 2 | 95 ± 15 | 120 ± 25 | 74 ± 16 | 2118 ± 249 | 3030 ± 565 | 15 ± 3 | 23 ± 4 | 26 ± 7 |
| Hs_37°C | 4 ± 2 | 84 ± 15 | 132 ± 12 | 388 ± 107 | 2397 ± 717 | 1837 ± 367 | 118 ± 72 | 30 ± 13 | 14 ± 2 |
| Hs_37°C + PGA1 | 4 ± 1 | 83 ± 34 | 134 ± 25 | 220 ± 47 | 2863 ± 813 | 2638 ± 530 | 64 ± 17 | 38 ± 15 | 20 ± 1 |

Ratios were calculated using the specific yield of BV-genomes and PFU (per cell), normalized to total cells per ml.

The mean fluorescence intensity measured by flow cytometry is detailed in Fig 3A and 3B. The mRFP fluorescence intensity was about one order of magnitude lower than that of EGFP. Different scales were used for each fluorophore to analyze the kinetics and not the fluorescence intensity. As expected, mRFP fluorescence increased before that of EGFP. However, both reporters require time to accumulate to be distinguishable from background, which happens between 24 hpi to 48 hpi. In this analysis, the peak in fluorescence for both reporters and from almost all conditions was observed at 72 hpi. The condition with the highest fluorescence was the same for both reporters, Hs_37°C + PGA1, but it was not significantly different from fluorescence levels in control condition.

At 72 hpi also, mRFP intensity was significantly different, with a lower mRFP fluorescence in Hs_constant_30°C, Hs_constant_30°C + PGA1, and Hs_30°C. At 96 hpi, fluorescence decreased in all conditions, and we found significant differences between various treatments and the control (Table 1 in S3 File). mRFP fluorescence in 27°C + PGA1 did not decrease as did fluorescence in all other treatments.

Interestingly, Hs_constant_30°C had a distinct pattern of EGFP expression, with its highest fluorescence levels at 48 hpi instead of 72 hpi, 24 hours earlier than what was observed under all other conditions. Moreover, at 72 hpi and 96 hpi, EGFP fluorescence in both Hs_constant_30°C and Hs_constant_30°C + PGA1 was among the lowest. These distinct alterations in each fluorescent reporter under certain conditions highlight the utility of DRBac for monitoring the activity of different promoters and reporters over time.

We investigated the correlation between mRFP fluorescence and PFU production, given that the baculovirus gene under the *vp39* promoter encodes VP39, the major nucleocapsid protein, essential for baculovirus nucleocapsid assembly. Analyzing the average PFU production and mRFP fluorescence across all eight conditions, we observed a correlation only at 24 hpi (r = 0.92, p = 0.0013) (Table 2 in S3 File). Interestingly, a correlation was also found between EGFP fluorescence and PFU production only at 24 hpi (r = 0.96, p = 0.0002). These results suggest that the observed correlations reflect overall baculovirus gene expression activity at this stage, rather than the specific expression of mRFP under the *vp39* promoter. Notably, the lack of correlation between BV-genomes and PFU or reporter's fluorescence at any time point, suggests that these relationships are not simply a reflection of overall baculovirus activity, but rather stage-specific processes influencing PFU production and reporter expression at 24 hpi.

Taking advantage of the individual cell measurements that flow cytometry performs, we made a second analysis for fluorescent reporter's signal only for [mRFP + /EGFP+] cells, with

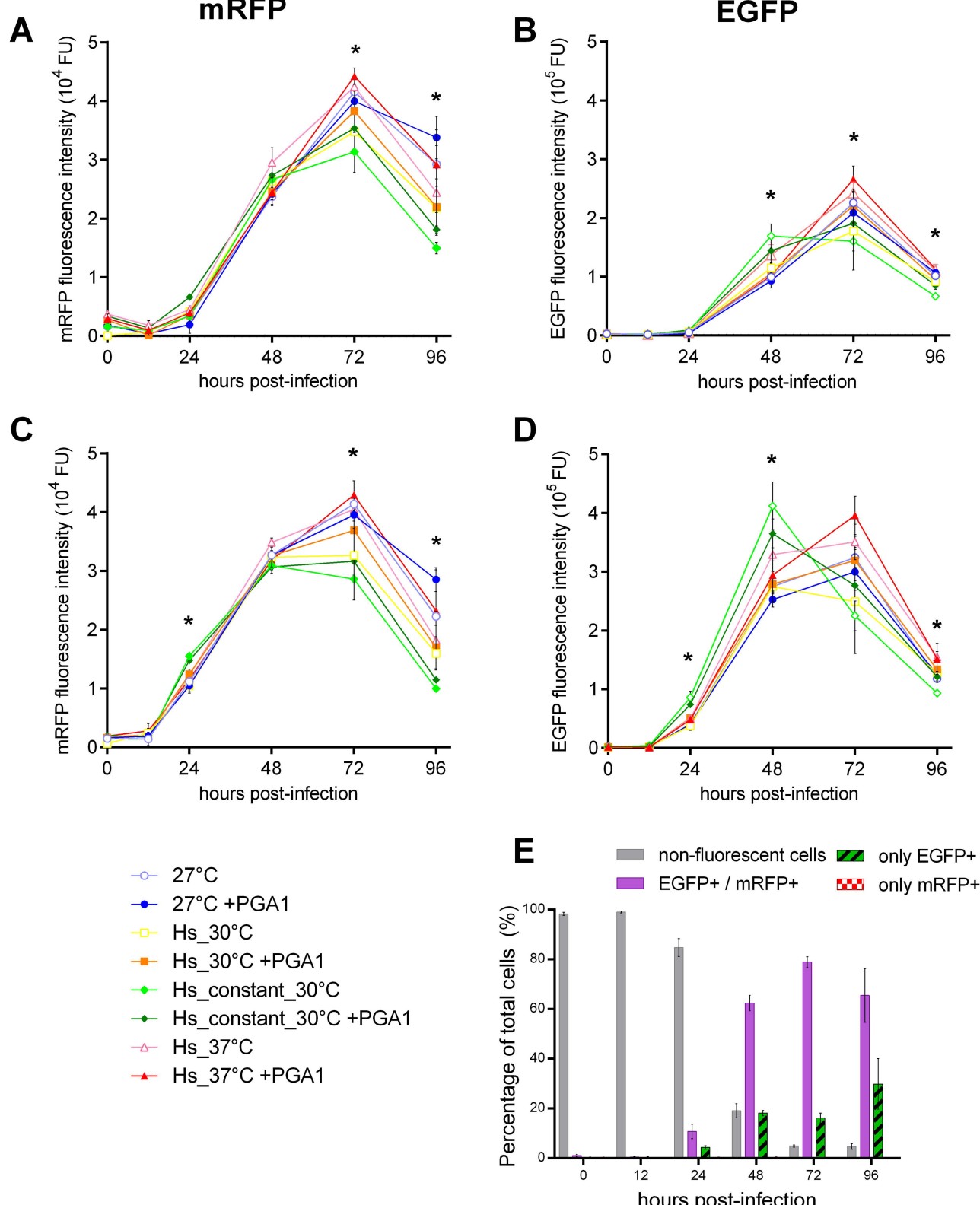

**Fig 3. Dynamics of EGFP and mRFP fluorescence in infected cells under different heat shock conditions.** Fluorescence of reporter proteins was measured by flow cytometry; note the different scales used for mRFP and EGFP fluorescence intensities. (A) and (B) show the geometric mean of cellular fluorescence intensity for mRFP and EGFP, respectively. (C) and (D) show the geometric mean fluorescence intensity for mRFP and EGFP only

in cells [mRFP + /EGFP+], the most abundant group. (E) Average kinetics of the distribution of cells according to their fluorescence profile. Asterisks indicate significant differences compared to the control for at least one condition at the indicated time points (p ≤ 0.05), determined by two-factor ANOVA with Tukey's multiple comparisons. The fluorescence profiles for the [mRFP − /EGFP+] and [mRFP + /EGFP−] groups are provided in Fig 2 in S3 File, and details of statistically significant differences from the control conditions fluorescence are in Table 1 in S3 File.

the geometric mean for fluorescence intensity shown in Fig 4, panels C and D. The differences previously observed are more obvious in this group in comparison to the mean values of all cells (Fig 3A, 3B). EGFP fluorescence can also be detected at 24 hpi, with low levels that increased until 72 hpi, to then decay at 96 hpi. We found significant differences from control EGFP fluorescence values at 24, 48 and 96 hpi. The EGFP fluorescence intensity in [EGFP + /mRFP+] cells under the Hs_constant_30°C condition was 1.49 times higher than in the same group at the control condition (27°C) at 48 hpi, and 1.17 times higher than the

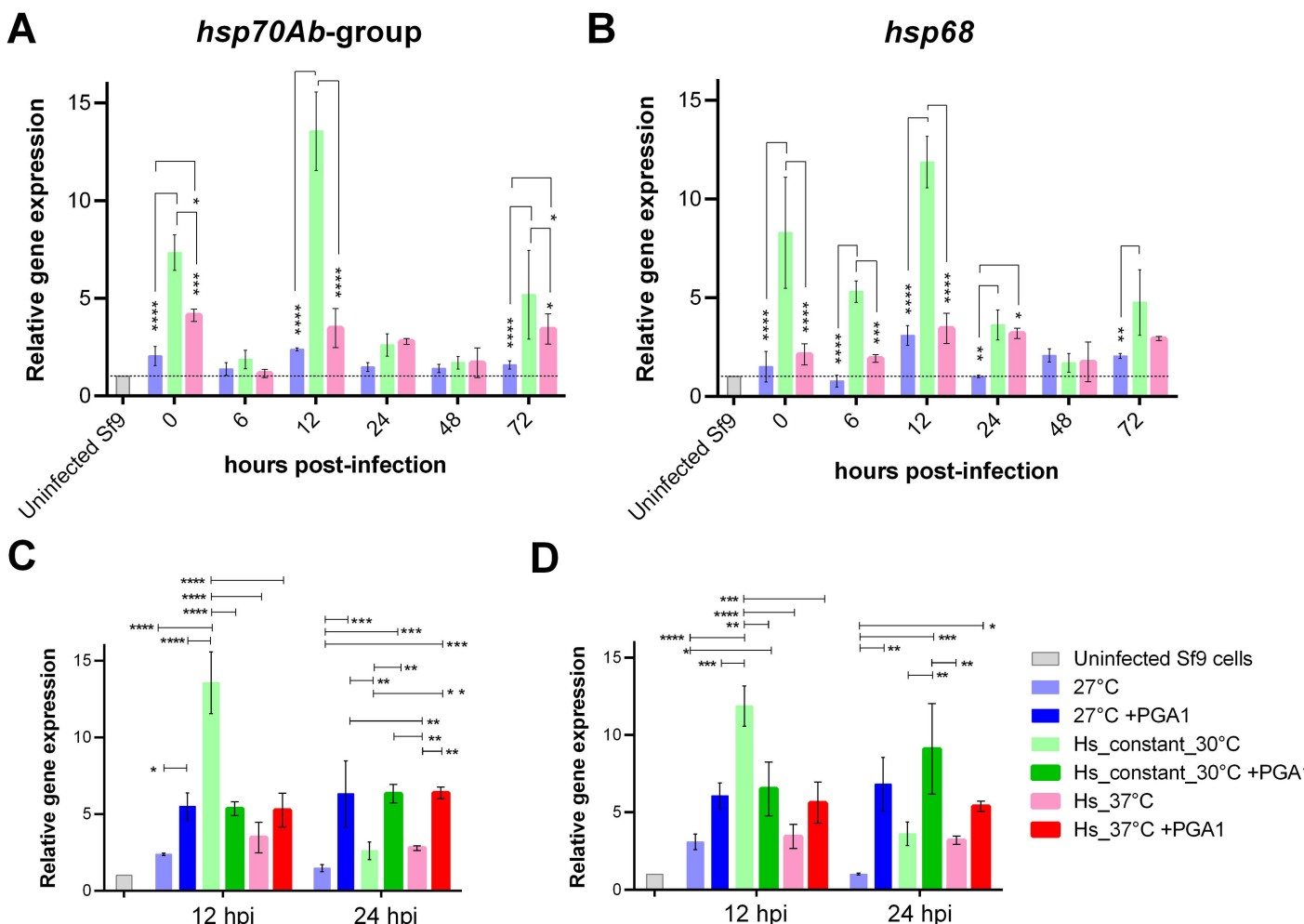

**Fig 4. Gene expression of the *hsp70Ab*-group and *hsp68* genes after baculovirus infection under different heat shock stimuli.** (A) and (B) show the expression profile of *hsp70Ab*-group and *hsp68,* respectively, at control and two selected heat shock conditions. The internal reference used for relative gene expression was the 28s gene. Uninfected Sf9 cells (gray bar) were analyzed in each run as a control to normalize and compare among different experiments, the dashed line indicates their expression level. (C) and (D) present the expression of the *hsp70Ab*-group and *hsp68* genes, respectively, for cells at 12 and 24 hours post-infection (hpi), under conditions with and without prostaglandin A1 (PGA1). Means and standard deviations are shown. ANOVA with two factors and Tukey's multiple comparisons were used to compare values among conditions at each time point. Significant differences are indicated by connecting lines and asterisks (p: * ≤ 0.05, ** ≤ 0.01, *** ≤ 0.001, **** ≤ 0.0001).

control condition at its highest point at 72 hpi. This suggests that under the condition of continuous 30°C stress, the infection cycle and productivity of the BEVS accelerate, reaching a fluorescence intensity similar or slightly superior 24 hours earlier than the control condition. It remains to be evaluated wether these improvements also occur with other recombinant proteins. The maximum fluorescence value for EGFP in the [mRFP + /EGFP+] group of cells was observed at 72 hpi under the Hs_37°C + PGA1 condition, followed by the same condition without PGA1 (Hs_37°C), representing 1.22 and 1.08 times the maximum fluorescence in the control condition, respectively.

We were interested in monitoring the dynamics in the expression of reporters and the proportion of cells that express each one, since variations may implicate differences in the infection dynamic or the baculovirus cycle among conditions. Using flow cytometry, cells were categorized into four groups based on the absence or detection of fluorescence from the two reporter proteins: only mRFP [mRFP + /EGFP−], only EGFP [mRFP − /EGFP+], both [mRFP + /EGFP+], or none [mRFP − /EGFP−]. The changes in the proportions of each group during the infection assay are depicted in Fig 3E. Typical data are shown, as no difference was observed in the distribution of the groups with any treatment. This suggests that the different heat shock treatments do not affect infection dynamics under these conditions that allow secondary infections (MOI =  1 PFU cell$^{-1}$).

At 24 hpi, the percentage of fluorescent cells for any reporter protein was low, with only $15.4 \pm 4.0\%$ on average across all conditions. Most cells fluoresced at 48 hpi ($80.9 \pm 5.3\%$), following the kinetics expected for a MOI of 1 PFU cell$^{-1}$. At 72 hpi, $95.0 \pm 4.9\%$ of cells were fluorescent. Cells that only expressed mRFP represented a minimal fraction, with less than 1% of cells. Most of the population were positive for both mRFP and EGFP fluorescence, while a subpopulation of cells only showed EGFP. The [EGFP + /mRFP+] group had the highest fluorescence intensity and the largest population fraction of fluorescent cells, with $78.8 \pm 2.6\%$ at 72 hpi. This group showed fluorescence intensity levels about 2 and 3 times higher than those expressing only EGFP or mRFP, respectively (Fig 2 in S3 File).

## Identification and complete sequencing of two *hsp70* genes responsive to heat shock and AcMNPV infection

To further investigate the cause of the differences observed in cells under various heat shock conditions, the expression of two hsp70 genes was measured. Two *hsp70* genes were of special interest since they were among several overexpressed host genes during an AcMNPV infection of *S. frugiperda* cells [18]. These genes were categorized as members of the HSP70 family by their partial sequence from expressed sequences tags (ESTs) used for microarray assay probes [18], but their complete coding sequence (CDS) and identification were unknown at the beginning of this project. Using alignments to other ESTs [31], and more recent transcriptomic data available for *S. frugiperda* in NCBI, we were able to design primers to amplify two complete CDS. Using PCR on cDNA from heat-shocked Sf9 cells, we successfully amplified and cloned these genes in a vector and obtained their complete sequence. The genes were named *hsp70Ab* and *hsp68* based on their similarity to *Drosophila melanogaster* genes and are now available in GenBank under accession numbers PQ108612 and PQ108613, respectively. Identifying these specific Hsp70 genes allows for more focused research into their roles and facilitates better analysis of existing studies. Their differentiation also helps clarify the distinct functions, expression profiles, and responses that various HSP70 family members may have during baculovirus infection.

We followed the relative gene expression of *hsp68* and *hsp70Ab* by reverse transcription qPCR (RT-qPCR) to monitor the effect of baculovirus infection combined with heat shock

stimuli on the cellular stress response. Given the presence of multiple nearly identical genes and isoforms in GenBank related to the *hsp70Ab* sequence, we designed primers targeting an identical region in these entries, named the *hsp70Ab*-group (details in S1 File). We assessed gene expression under three conditions: Hs_constant_30°C, Hs_37°C, and control (27°C) (Fig 4A, 4B), with or without PGA1, selected for their significant differences in viral titer (PFU), BV-genomes, and fluorescence. A statistical comparison was conducted among the expression levels in each condition. Significant differences were seen at 0, 6, 12, 24, and 72 hpi, as depicted in Fig 4.

In the control condition (infected and maintained at 27°C), only the expression of *hsp68* slightly increased at 12 hpi. The heat shock conditions tested induced higher expression of both genes but with different expression profiles. Notably, the expression of the *hsp70Ab*-group was approximately $6.4 \pm 1$ times higher than that of *hsp68*, likely in part due to the multiple genes and isoforms targeted by the *hsp70Ab*-group primers. The expression of both genes remained above baseline levels of uninfected Sf9 cells throughout infection (Fig 4A, 4B). Interestingly, at 0 hpi, the Hs_constant_30°C condition resulted in higher heat shock gene expression of both genes, compared to the control and Hs_37°C, despite the latter representing a more intense heat shock. This likely reflects that 37°C is too high a temperature, impairing the cellular stress response.

This condition, Hs_constant_30°C, showed the highest levels of expression in both HSP70 genes, with a peak level at 12 hpi and additional peaks at 0 and 72 hpi for the *hsp70Ab*-group, and at 0, 6, 24, and 72 hpi for *hsp68*. Previous studies have reported a peak in the expression of individual Hsc70 and Hsp70 genes at 6 hpi under baculovirus infection, followed by a marked decline after this time, reaching the lowest levels at 24 hpi [17,18]. The differences in expression kinetics observed here may be attributed to the lower multiplicity of infection (MOI) used in our experiments (1 PFU cell$^{-1}$), compared to the higher MOI of 10 PFU cell$^{-1}$ used in those studies, and our combination of heat shock stimuli with the baculovirus infection.

The Hs_constant_30°C condition resulted in the highest PFU production and gene expression levels at 24 hpi, but the lowest PFU titers at 72 hpi (Fig 2). EGFP fluorescence intensity followed a similar trend under this condition, with the highest values among all conditions at 24 and 48 hpi, then dropping to the lowest levels at 72 hpi (Fig 3D). Notably, although a secondary peak in gene expression was observed at 72 hpi, this did not correspond to an increase in PFU production or fluorescence intensity. As mentioned in the Introduction section, the role of Hsp70 genes during baculovirus infection has been linked mainly to the first phases of infection, which aligns with our observations, where the highest expression levels of both genes were detected at 12 hpi under all conditions, and the majority of differences are within the 6 to 24 hpi timeframe.

We conducted exploratory correlation analyses between the expression of the HSP70 and productivity metrics (PFU production, BV-genomes, mRFP and EGFP fluorescence) to investigate potential trends. At 24 hpi, the *hsp70Ab*-group showed moderate positive correlations with PFU, BV-genomes, and fluorescence (0.87, 0.75, and 0.63, respectively), while *hsp68* exhibited stronger correlations of 0.97, 0.54, and 0.82. Interestingly, by 48 hpi, correlations for the *hsp70Ab*-group strengthened to 0.99, 0.90, and 0.92, whereas *hsp68* shifted dramatically to strong negative correlations of $-0.98$, $-0.94$, and $-0.96$. At 72 hpi, *hsp70Ab*-group correlations decreased to -0.80, -0.03, and -0.60, while *hsp68* correlations shifted to $-0.90$, 0.17, and -0.75. These patterns suggest a complex temporal relationship between these HSP70 genes expression and productivity metrics. Notably, the Hs_constant_30°C condition showed the highest expression profile among all conditions for both *hsp70Ab*-group and *hsp68*, with significative differences at 0, 6, 12, 24 and 72 hpi. Under this condition we also found the highest PFU production and fluorescence values for both reporters only at 24 hpi, and for EGFP at 48 hpi.

While we did not observe a significant effect of PGA1 in productivity, we examined its impact on the induction of the selected *hsp70* genes under the three selected conditions at 12 and 24 hpi, when the largest differences in titer and BV-genomes production were observed. We found a significant effect of PGA1 on the expression of both genes. Interestingly, PGA1 addition resulted in similar expression levels in the control and heat-shock conditions (Fig 4C, 4D). On average, *hsp70Ab*-group expression increased 2.25-fold and *hsp68* 1.96-fold compared to control conditions levels of expression without PGA1 at 12 hpi. At 24 hpi, we observed a higher difference with a six-fold increase for *hsp70Ab*-group, and a seven-fold difference for *hsp68*, compared to the control without PGA1 at 24 hpi. Despite this strong induction, the lack of variation in *hsp70* genes expression between PGA1-treated conditions contrasts with the observed differences in productivity metrics under heat shock conditions. Interestingly, we did not observe the same high levels of induction in both genes under the Hs_constant_30°C condition with and without PGA1 at 12. We suspect that a generalized higher induction of HSP70 genes by PGA1 may reduce, impair or just make it unnecessary the induction of specific HSP70 genes by other mechanisms.

In general, these observations highlight that certain heat shock conditions may enhance both HSP70 expression and productivity at specific time points. However, the temporal trends suggest that any potential relationship between HSP70 expression and productivity is dynamic and may vary depending on the stage of infection. Previous studies have reported that inhibiting HSP70 proteins early in infection reduces viral DNA replication and titer production with less pronounced effects when inhibition occurred later in the infection [19]. While our results showed stronger correlations between HSP70 genes expression and productivity metrics at 24 hpi, the only temporal coincidence and the contrasting results under PGA1 conditions suggest that HSP70 overexpression may not be directly related to productivity outcomes. However, HSP70 expression may still contribute to these differences, as seen in the Hs_constant_30°C condition, where higher HSP70 expression coincided with increased productivity metrics at specific time points. This suggests that the relationship could be indirect, context-dependent, and influenced by the baculovirus infection activity and timing. Other factors, altered by heat shock but not PGA1, beyond the HSP70 genes overexpression, may therefore contribute to the observed variations in productivity under these conditions.

## Discussion

In this study, we evaluated the baculovirus infection cycle development and productivity under different heat shock stimuli combined with PGA1 addition. We discovered that two of the three heat shock conditions enhance the production of the fluorescent reporters associated with promoters of the late and very-late phases of the viral cycle or increase the BV titers or BV-genomes but only at certain times post-infection. The impact of heat shock on the progression of the baculovirus life cycle, the balance between BV progeny, the relation between BV-genomes and productive BV, and the expression of heat shock genes had not been reported before this work, even when a relation between heat shock gene expression and baculovirus infection of *S. frugiperda* cells had been previously suggested. It was found that different heat shock stimuli have distinct effects in the early and late infection phases. These results highlight the intricate relationship between stress responses and viral or recombinant protein production. Also, the complete CDS and the identification of relevant inducible heat shock genes were partially available before, in this study they were identified and completely sequenced. This information allows for a better understanding of the role of heat shock in baculovirus life cycle progression, along with identifying the conditions that result in higher baculovirus or recombinant protein productivity, the main products of the BEVS.

We observed that constant heat shock at 30°C led to an initial increase in PFU at 24 hpi, suggesting that moderate stress can accelerate early viral production. However, this same condition resulted in the lowest PFU levels at 72 hpi, indicating a potential trade-off between early gains in productivity and long-term cellular functionality. A similar behavior was observed with the EGFP reporter under the very late *polh* promoter, where fluorescence was the highest at 24 and 48 hpi, but declined significantly at 72 and 96 hpi. Other stress conditions (Hs_37°C) dramatically increased BV-genomes at 24 hpi without an equivalent increase in the number of functional PFU. These findings suggest that, while heat shock can enhance early-phase production, prolonged exposure may induce cellular exhaustion or dysregulation of key processes, ultimately impairing later-stage productivity.

The differential responses observed under various heat shock conditions highlight the complex role of Hsp70 proteins in the baculovirus infection cycle. The specific expression patterns of the sequenced *hsp68* and *hsp70Ab* genes suggest that these proteins play distinct roles in managing cellular stress and supporting viral replication. Previous studies have shown that Hsp70 proteins are involved in critical processes such as viral protein transport, the stability of viral transcription factors, and genome replication at early times of baculovirus infection [15,16,19,20]. However, our data also point out that excessive or prolonged stress and expression of these genes can have deleterious effects, potentially leading to a loss of cellular integrity and reduced viral yields in the later stages of infection. It is important to consider that while chaperone proteins, like Hsp70, may have beneficial effects under stress conditions, their production is costly to the cell and can have negative effects [34].

Several studies have demonstrated that co-expression of molecular chaperones, including proteins other than Hsp70, can improve the quantity, proper folding, and solubility of recombinant proteins in the BEVS [35–42]. Interestingly, although Hsp70 proteins are known to enhance the solubility and stability of recombinant proteins, our experiments revealed only a temporal correlation, observed at 24 hpi, between HSP70 genes expression levels and the fluorescence of recombinant reporters or PFU titers. This discrepancy suggests that other mechanisms or interacting factors that take place in response to heat shock are involved but were not directly assessed in this study. To delve deeper into the levels of expression of specific *hsp70* genes, we monitored the expression of two previously associated genes with up-regulation during a baculovirus infection in *S. frugiperda* cells: *hsp68* and *hsp70Ab*-group. We found that each gene or group of genes has a different expression profile under different heat shocks, with a peak at 12 hpi using a MOI of 1. Further studies are required to evaluate the relation of the changes observed in the expression of these genes and the observed changes in productivity of the system and evaluate which other factors or genes may be altering the productivity of the BEVS after the heat shock stimuli to the host cells.

For conditions combined with PGA1, we aimed to increase the cellular heat shock response without increasing the temperature or duration of the heat shock exposure. This approach would highlight the possible positive effects of the heat shock response of insect cells without an excess of the negative effects of stress. However, we did not observe significant differences in titer, BV-genomes, or fluorescence production under PGA1 treatment. When we evaluated its effect on the induction of *hsp68* or *hsp70Ab*-group genes at 12 or 24 hpi, we found significant increases in all PGA1-treated conditions with transcription levels reaching similar values regardless of the presence of heat shock. These results indicate that the main driver of productivity differences among conditions was the heat shock stimulus, rather than a direct effect of *hsp70s* expression levels. Nonetheless, the temporal patterns observed in our data suggest that HSP70 genes may have a more pronounced influence during early stages of infection, when its expression levels peak.

At 72 hpi, we noticed a $21 \pm 7\%$ increase in EGFP fluorescence in all conditions that received a heat shock combined with PGA1 ([EGFP+/mRFP+] group of cells), an effect not observed in the control condition with PGA1 (27°C + PGA1). Taken together, these findings suggest that the cellular stress response, including increased expression of *hsp70* genes and likely changes in other stress-related genes, may enhance BEVS productivity during defined stages of the infection cycle, such as the early and late phases when viral replication is initiated and BV production starts, and the late phase when recombinant protein expression peaks.

In conclusion, our study underscores the importance of the host stress response in optimizing baculovirus-based production systems. The ability to precisely modulate the host's stress response, particularly through controlled expression of Hsp70 proteins, could be key to improving the efficiency and robustness of BEVS for a wide range of biotechnological applications. Given these findings, future research should focus on strategies to balance stress induction with productivity enhancement in the BEVS. Promising avenues are the targeted modification or overexpression of distinct *S. frugiperda hsp70* genes, and optimization of their induction times over the viral cycle to increase either BV titers or recombinant proteins. This approach could improve not only viral progeny production but also the quality and yield of recombinant proteins. Positive results have been seen with the co-expression of heterologous Hsp70 chaperons in the BEVS system [42]. We propose that the endogenous Hsp70 genes of *S. frugiperda* may have distinct effects compared to recombinant ones and that each *S. frugiperda* HSP70 gene may have unique roles and impact, warranting additional research. Additional factors, such as cochaperones like *hsp40*, can also play a significant role in directing the activity of other Hsp70 proteins. Furthermore, exploring the timing and intensity of stress preconditioning (e.g., heat shock) offers a promising avenue for enhancing BEVS efficiency across various proteins, host systems, conditions, and production scales.

## Supporting information

**S1 File. Primers design and HSP70 genes sequence analysis.**
(PDF)

**S2 File. Cell culture concentrations, viability assays, and information on MOI 5 infections.**
(PDF)

**S3 File. Flow cytometry analysis and statistical evaluation of fluorescence differences for each reporter.**
(PDF)

## Acknowledgments

Technical support by Vanessa Hernández, Martha Contreras, Mabel Rodríguez, and Marisela Olvera. Enrique Paz Cortés is a doctoral student from the Programa de Doctorado en Ciencias Biomédicas, Universidad Nacional Autónoma de México (UNAM).

## Author contributions

**Conceptualization:** Enrique Paz-Cortés, Laura A. Palomares.

**Data curation:** Enrique Paz-Cortés.

**Formal analysis:** Enrique Paz-Cortés.

**Funding acquisition:** Laura A Palomares.

**Investigation:** Enrique Paz-Cortés, Ana Ruth Pastor, Laura A. Palomares.

**Methodology:** Enrique Paz-Cortés, Ana Ruth Pastor, Roberta Salinas-Marín, Laura A. Palomares.

**Project administration:** Ana Ruth Pastor, Octavio T. Ramírez, Laura A. Palomares.

**Resources:** Roberta Salinas-Marín, Octavio T. Ramírez, Laura A. Palomares.

**Supervision:** Ana Ruth Pastor, Laura A. Palomares.

**Writing – original draft:** Enrique Paz-Cortés, Laura A. Palomares.

**Writing – review & editing:** Enrique Paz-Cortés, Octavio T. Ramírez, Laura A. Palomares.

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
