## [Decision Letter · Decision Letter 0]

15 Dec 2024

PONE-D-24-50172Molecular Characterization of the Effects of Heat Shock on the Infection Cycle Progression and Productivity of the Baculovirus Expression Vector SystemPLOS ONE

Dear Dr. Palomares,

Thank you for submitting your manuscript to PLOS ONE. After careful consideration, we feel that it has merit but does not fully meet PLOS ONE’s publication criteria as it currently stands. Therefore, we invite you to submit a revised version of the manuscript that addresses the points raised during the review process.

We look forward to receiving your revised manuscript.

Kind regards,

Jian Xu, Ph.D.

Academic Editor

PLOS ONE

Journal Requirements: When submitting your revision, we need you to address these additional requirements. 1. Please ensure that your manuscript meets PLOS ONE's style requirements, including those for file naming. The PLOS ONE style templates can be found at https://journals.plos.org/plosone/s/file?id=wjVg/PLOSOne_formatting_sample_main_body.pdf and https://journals.plos.org/plosone/s/file?id=ba62/PLOSOne_formatting_sample_title_authors_affiliations.pdf 2. We note that the grant information you provided in the ‘Funding Information’ and ‘Financial Disclosure’ sections do not match.  When you resubmit, please ensure that you provide the correct grant numbers for the awards you received for your study in the ‘Funding Information’ section. 3. Thank you for stating the following financial disclosure: "Universidad Nacional Autónoma de MéxicoDGAPA-UNAM PAPIIT IT200824Enrique Paz Cortés received a fellowship from CONAHCYT 270335." Please state what role the funders took in the study.  If the funders had no role, please state: ""The funders had no role in study design, data collection and analysis, decision to publish, or preparation of the manuscript."" If this statement is not correct you must amend it as needed. Please include this amended Role of Funder statement in your cover letter; we will change the online submission form on your behalf. 4. Thank you for stating the following in the Acknowledgments Section of your manuscript: "Technical support by Vanessa Hernández, Martha Contreras, Mabel Rodríguez, and Marisela Olvera. Funding by UNAM and DGAPA-UNAM PAPIIT IT202824. Enrique Paz Cortés is a doctoral student from the Programa de Doctorado en Ciencias Biomédicas, Universidad Nacional Autónoma de México (UNAM) and received CONAHCYT fellowship 270335." We note that you have provided funding information that is not currently declared in your Funding Statement. However, funding information should not appear in the Acknowledgments section or other areas of your manuscript. We will only publish funding information present in the Funding Statement section of the online submission form. Please remove any funding-related text from the manuscript and let us know how you would like to update your Funding Statement. Currently, your Funding Statement reads as follows: "Universidad Nacional Autónoma de MéxicoDGAPA-UNAM PAPIIT IT200824Enrique Paz Cortés received a fellowship from CONAHCYT 270335." Please include your amended statements within your cover letter; we will change the online submission form on your behalf.

Reviewers' comments:

Reviewer's Responses to Questions

**Comments to the Author**

1. Is the manuscript technically sound, and do the data support the conclusions?

Reviewer #1: Partly

Reviewer #2: Yes

2. Has the statistical analysis been performed appropriately and rigorously? 

Reviewer #1: Yes

Reviewer #2: Yes

3. Have the authors made all data underlying the findings in their manuscript fully available?

Reviewer #1: Yes

Reviewer #2: Yes

4. Is the manuscript presented in an intelligible fashion and written in standard English?

Reviewer #1: Yes

Reviewer #2: Yes

5. Review Comments to the Author

Reviewer #1: Abstract: the wording used in the abstract suggests that wild type AcMNPV has been used - it should be made clear that recombinant AcMNPV expressing reported genes was used. The accepted abbreviation for the virus should be AcMNPV.

Introduction: The replication cycle of AcMNPV is very much over simplified and inaccurate. The reader is led to understand that the four stages of the virus replication cycle have hard cut offs between the very early, early, late and very late stages. At several points in the text and in Fig 1, statements such as BV stops in the VL phase, BV starts in the L phase, polyhedrin protein/polh promoter is made/is active in the VL phase. There are no hard stops between the phases, BV starts being made from 15 hours and continues into the VL phase but starts to decline. The polh promoter starts about 15-18 hpi and builds through the L phase with a boost in the VL phase - recombinant proteins can be detected from about 18 hpi. The introduction should be redrafted to more accurately reflect the AcMNPV replication cycle.

Further in the introduction, it is stated that Sf is a primary host for AcMNPV - it is not. Sf is not a natural host for AcMNPV but the virus replicates to unusally high titres in the Sf9 cell line, which is why the BEVS system uses Sf9 cells to produce high titre stocks of recombinant viruses. If you take a natural host cell, such as Tni (e.g. Hi5 cell line), then the virus does not replicate to such high titres but the polh promoter is much more active - giving generally higher recombinant protein yields, which is why many use Tni cells for recombinant protein production (but always Sf cells for virus production). So really this study has not been carried out in the 'natural' host for AcMNPV, however, the study can be justified because Sf cells are the cells used for the BEVS system - the introduction should be updated to reflect this.

Final point, I don't think it is 'crucial' to the development of BEVS that we know the effect of heat shock proteins, it is helpful or useful but not crucial.

Materials and Methods: In the results, figures and text refer to PFU of virus but in the methods, TCID50 is used to titrate viruses - please state how PFU was redrived fro TCID50.

Results: In 3.1, the results are described but I think a key result has been overlooked namely that the heatshock for constant 30C results in a timing difference for BV production compared to the 27 C control - at 30 C BV peaks at 48 and then declines where at 27 C, BV continues to increase until the last time point sampled at 72. It may have been helpful in this study to have gone beyond 72 h to see when the controls peaks (this may be a consequence of the authors thinking of the replication cycle in discrete phases rather than in a continuous cycle).

Later results do support the initial data in 3.1 that continuous heat shock at 30 results in timing differences (e.g. Fig 3).

Coexpression of two reporter genes demonstated a clear shift in the timing of expression, to earlier in the replication cycle but I could not see clear evidence of an increase in expression overall. Therefore the benefits of heat shock proteins may be beneficial to certain types of recombinant proteins, e.g. those that need to be processed (such as glycoproteins or secreted proteins) before the cell machinery becomes compromised through virus infection. Such observations have been made before and are the basis of a number of modified Sf9 cell lines stably expressing genes that aid in the expression of some proteins. I can't see this being referenced in the discussion. I think based on the results presented, the discussion and conclusion need to be modified to be more specific in the claims about the benefits of coexpressing/inducing heatshock proteins.

The paper is very readable and the figures well presented. Overall, once the concerns above have been addressed I think this will make a very useful contribution to those working to improve the BEVS system.

Reviewer #2: Molecular Characterization of the Effects of Heat Shock on the Infection Cycle Progression and Productivity of the Baculovirus Expression Vector System.

In this work, the authors evaluated the baculovirus infection cycle development and productivity at low MOI and under different heat shock stimuli combined with PGA1 addition.

Please find the comments below:

MAJOR COMMENTS

Relevance of the study using a low MOI.

We understand that a low MOI infection refers to a two-step baculovirus infection process where cells are inoculated with an MOI < 1, leading to only partial infection of the cell culture. In this scenario, uninfected cells continue to proliferate, while infected cells produce recombinant protein and new virus particles, necessitating a second round of infection. This approach results in longer processing times but often achieves higher protein expression yields, as more cells are available to produce the recombinant protein.

However, most research groups typically use a high MOI (5–10) to shorten production time, utilising a higher TOI to enhance expression. It would be valuable to understand how transferable the findings are when using a high MOI, as this aspect was not thoroughly addressed in the manuscript (it is briefly mentioned in comparisons with other labs).

Have the authors conducted any heat shock treatments with high MOI and TOI? If so, including this data in the manuscript would strengthen its applicability and understanding. I'd suggest narrowing the focus to one or two optimal conditions and performing PFU/mL and BV-genomes/mL measurements for clarity. Comparing both MOI strategies could bring an overview of the molecular characterisation of the infection process.

PFU and BV-genome per cell.

Table 2 presents PFU and BV-genomes per cell. However, as the authors noted, not all cells will be infected, making these values unreliable when quantified per cell. Expressing the units per volume would be more appropriate, ensuring consistency. The final ratio (BV-genomes to PFU) can still be provided if it refers to the same sample volume.

vp39 and p10 promoters.

As the authors noted, there are distinct infection phases during the baculovirus lytic cycle. The absence of cells expressing only RFP suggests a narrow time window between the expression driven by the selected late and very late promoters (vp39 and p10). This overlap makes it challenging to differentiate between infection stages, particularly in an asynchronous infection. To address this, the authors may want to consider performing a high MOI infection to synchronize expression and potentially identify time points where only RFP is expressed, as higher cell density would be achieved. Otherwise, using two different reporter genes may not be necessary, as it does not provide additional information. This is evident in Figure 3, where the temporal overlap of both reporter genes is clear.

Related to the reporters, in Section 3.3, it would be helpful to address the observed discrepancy between higher PFU values and lower fluorescence levels of mRFP.

Hsp70 levels.

Analysing Hsp70 levels in a heterogeneous population (<10% infected cells) may not provide a representative sample to establish clear trends. As the authors noted, it is intriguing that the expression levels are high at 0 h post-infection (likely due to the heat shock applied hours before infection) but decrease at 6 hpi before rising again at 12 hpi. While the difference is evident compared to the control (infected cells at 27°C) is significant at 12 hpi, the observed trend remains difficult to interpret. Could the authors provide additional data or reference literature to clarify this phenomenon?

Additionally, in Figure 4, panels C and D, only the relative expression levels at 24 hpi are shown. It would be valuable to include data for 12 hpi as well to confirm the effect of PGA1 addition on expression trends.

Regarding the statement, “The differences observed in titers or BV-genomes among conditions at 24 hpi indicate that the variation in productivity observed is driven by factors beyond or in addition to the altered expression of the hsp70 genes analyzed”: we recognise that the regulation of the hsp70 family is highly complex and challenging to interpret. However, based on the data provided, it is not possible to demonstrate a correlation between the expression profiles under different heat shock conditions and the observed changes in productivity (measured as reporter gene expression). Both effects appear to result from the heat shock but may not necessarily influence one another directly to produce an integrated outcome.

MINOR COMMENTS:

Figure 1

Since the results are based on the infection timeline, I suggest including the hours post-infection in the figure's cycle. This addition will provide clarity on why specific time points were chosen for the time course experiments and will emphasise the differences between the primary and secondary infection cycles.

Colour Usage

While I understand that green and red are commonly used to represent EGFP and RFP, these colours are not suitable for individuals with colour blindness. I recommend considering an alternative colour scheme or employing distinct patterns to make the figure more accessible to all readers.

Fig3 AND SupInf2.

The graduations on the X-axis are not evenly spaced, which is essential for this type of graph (as it is not a histogram). I recommend ensuring that the intervals are consistent to accurately represent the data and align with standard graphing conventions.

Fig. SI_2.1 AND Fig S3.2.

The graph legend should not be presented as a separate panel. I recommend integrating it directly into the graph for better readability and a more cohesive presentation.

6. PLOS authors have the option to publish the peer review history of their article (what does this mean? ). If published, this will include your full peer review and any attached files.

**Do you want your identity to be public for this peer review?** For information about this choice, including consent withdrawal, please see our Privacy Policy .

Reviewer #1: No

Reviewer #2: No

---

## [Author Response · Author response to Decision Letter 1]

5 Feb 2025

Molecular characterization of the effects of heat shock on the infection cycle progression and productivity of the baculovirus expression vector system

By Enrique Paz-Cortés et al.

Rebuttal letter

We thank the editors and reviewers for their constructive and careful manuscript revision. In this document, we present a point-by-point response to all comments. To facilitate reading, the original comments are in blue text below.

Reviewer # 1:

Comment 1 of 9 - Abstract: the wording used in the abstract suggests that wild type AcMNPV has been used - it should be made clear that recombinant AcMNPV expressing reported genes was used. The accepted abbreviation for the virus should be AcMNPV.

Thank you for this observation. We have clarified in the abstract that a recombinant AcMNPV expressing two reporter genes was used and ensured consistent use of the abbreviation AcMNPV throughout the manuscript

Comment 2 of 9 - Introduction: The replication cycle of AcMNPV is very much over simplified and inaccurate. The reader is led to understand that the four stages of the virus replication cycle have hard cut offs between the very early, early, late and very late stages. At several points in the text and in Fig 1, statements such as BV stops in the VL phase, BV starts in the L phase, polyhedrin protein/polh promoter is made/is active in the VL phase. There are no hard stops between the phases, BV starts being made from 15 hours and continues into the VL phase but starts to decline. The polh promoter starts about 15-18 hpi and builds through the L phase with a boost in the VL phase - recombinant proteins can be detected from about 18 hpi. The introduction should be redrafted to more accurately reflect the AcMNPV replication cycle.

We appreciate the reviewer’s comment regarding the oversimplification of the AcMNPV replication cycle in the original manuscript. We have revised the Introduction to provide a more accurate and detailed description of the replication cycle, emphasizing the continuity and overlap between the four phases.

Specifically, we have clarified that BV production begins during the late phase (approximately 15 hpi) and continues into the very late phase, although at reduced levels. Additionally, we have corrected the description of the polh promoter, noting that its activity begins around 15–18 hpi, with an increase in its expression at 24 hpi and peaks further during the very late phase. We added references to support the temporal division of the baculovirus cycle. We also modified the drawings and description in Figure 1 to remove the elements that suggest abrupt cut-offs in the baculovirus cycle. We believe these changes address the reviewer’s concerns and enhance the scientific accuracy of the manuscript.

Comment 3 of 9 - Further in the introduction, it is stated that Sf is a primary host for AcMNPV - it is not. Sf is not a natural host for AcMNPV but the virus replicates to unusually high titres in the Sf9 cell line, which is why the BEVS system uses Sf9 cells to produce high titre stocks of recombinant viruses. If you take a natural host cell, such as Tni (e.g. Hi5 cell line), then the virus does not replicate to such high titres but the polh promoter is much more active - giving generally higher recombinant protein yields, which is why many use Tni cells for recombinant protein production (but always Sf cells for virus production). So really this study has not been carried out in the 'natural' host for AcMNPV, however, the study can be justified because Sf cells are the cells used for the BEVS system - the introduction should be updated to reflect this.

We thank the reviewer for these insightful comments. We have revised the Introduction to clarify that Spodoptera frugiperda-derived cell lines, including Sf9, are among the most used cell lines for AcMNPV studies and are widely employed in BEVS for producing recombinant virus stocks. Additionally, we noted that Trichoplusia ni-derived cell lines, such as Hi5, are often preferred for recombinant protein production due to the stronger activity of the polh promoter in these cells.

We selected Sf9 cells for this study specifically because their ability to produce high viral titers aligns with our focus on investigating the effects of early HSP70 gene induction on the viral cycle. Previous studies have shown that early inhibition of HSP70 proteins significantly reduces viral DNA replication and BV production. This supports our hypothesis that earlier alterations (induction) in HSP70 protein levels may have a greater impact on viral titers and BV-genome production.

The manuscript has been updated to reflect these considerations and to provide more accurate context for the selection of Sf9 cells in this study. A reference has been added.

Comment 4 of 9 - Final point, I don't think it is 'crucial' to the development of BEVS that we know the effect of heat shock proteins, it is helpful or useful but not crucial.

We agree with your comment. We have revised the manuscript to replace the term "crucial" with "useful". This acknowledges that while understanding the effects of Hsp70 proteins is not indispensable, it provides valuable insights into the baculovirus-host cell relation, and we think our results can be useful to enhance the performance of the system.

Comment 5 of 9 - Materials and Methods: In the results, figures and text refer to PFU of virus but in the methods, TCID50 is used to titrate viruses - please state how PFU was redrived from TCID50.

We have updated the Materials and Methods section to explicitly state how PFU values were derived from TCID50. Specifically, the PFU/mL was calculated using the formula:

PFU/mL=TCID50 /mL×0.69, as described in (Mena et al., 2003). This reference also details the standard conditions and cell concentrations that we routinely use for seeding Sf9 cells in 96-well plates for tittering assays. Additionally, we have improved the information by including a mention of the Reed-Muench method, which was used to calculate the TCID50 values.

Comment 6 of 9 - Results: In 3.1, the results are described but I think a key result has been overlooked namely that the heat shock for constant 30C results in a timing difference for BV production compared to the 27 C control - at 30 C BV peaks at 48 and then declines where at 27 C, BV continues to increase until the last time point sampled at 72. It may have been helpful in this study to have gone beyond 72 h to see when the controls peaks (this may be a consequence of the authors thinking of the replication cycle in discrete phases rather than in a continuous cycle). Later results do support the initial data in 3.1 that continuous heat shock at 30 results in timing differences (e.g. Fig 3).

Thank you for your thoughtful feedback. We appreciate your observation regarding the timing differences in BV production under the constant 30°C heat shock condition compared to the 27°C control. Upon reviewing your comment and revisiting our data, we believe there may have been some misunderstanding regarding the results presented in Figure 2.

From our data, all conditions show a peak in BV production (both PFU and BV-genomes) at 72 hpi (except BV-genomes at Hs_37°C). For the constant_30°C conditions, there is still an increase in BV titer between 48 and 72 hpi, with the proportion of increase being just 1.1 and 1.3 times their levels at 48 hpi (without PGA1 and with PGA1, respectively). These proportions represent the lowest increments among the tested conditions, but they confirm a continuous increase from 48 to 72 hpi, rather than a decline as suggested in your comment. This is reflected also in the specific titers production in Table 2.

We routinely evaluate PFUs production up to 96 hpi every 24 hours, particularly in experiments conducted at MOI of 1 or lower. Extending the analysis to 96 hpi, we found only two conditions—Hs_constant_30°C and Hs_37°C +PGA1—showed further increments in PFUs (1.7- and 1.5-fold increases, respectively). However, BV-genome levels were not measured at 96 hpi due to a significant drop in cell viability (approximately 20% across conditions), which could compromise the reliability of the assay. This is why we did not present the comparison of both data at this time point. Additionally, we typically collect virus stocks before viability drops below 40% to avoid problems related to excess of cellular debris.

We acknowledge that viewing the viral replication cycle as a series of discrete phases may oversimplify its dynamics, potentially leading to inaccurate interpretations of the data. To address this, we have made revisions in the introduction, as suggested, to emphasize the continuous nature of the cycle. Additionally, we have included PFU titration data at 96 hpi as supplemental material (Fig 2 in S2 File) , recognizing the importance of this aspect highlighted by your comment. The supplemental materials now include a graphical representation of BV titration from 24 to 96 hpi, along with two tables: one showing the maximum production for each condition and the relative production proportions at each time point (Table 2 in S2 File), and another detailing the proportion of increase between consecutive sampling points (Table 3 in S2 File).

We appreciate your suggestion to extend the study beyond 72 hpi. While we have included data for PFU production at 96 hpi in this response, future experiments will aim to assess BV-genomes and other parameters at extended time points, provided cell viability remains sufficient for reliable measurements.

Comment 7 of 9 - Coexpression of two reporter genes demonstrated a clear shift in the timing of expression, to earlier in the replication cycle but I could not see clear evidence of an increase in expression overall. Therefore, the benefits of heat shock proteins may be beneficial to certain types of recombinant proteins, e.g. those that need to be processed (such as glycoproteins or secreted proteins) before the cell machinery becomes compromised through virus infection.

We appreciate your observation regarding the shift in timing of reporter gene expression and agree with your insights. This comment highlights three important aspects:

1) As you mentioned, the observed timing shift could be particularly beneficial to produce recombinant proteins that require processing (e.g., glycoproteins or secreted proteins) or are toxic to cells. Also, we think that it is worthy to keep researching the conditions that improved BV titters at earlier time points to discover elements that may be useful to improve the production of recombinant BV.

2) We also found it noteworthy that under certain heat shock conditions (e.g., Fig. 3D), EGFP expression peaked earlier. Specifically, the levels of EGFP expression at 48 hpi under the Hs_constant_30ºC condition were comparable to the maximum levels observed at 72 hpi in the control and other conditions. This earlier peak could have practical implications for bioprocesses where reducing production time is advantageous.

3) As future directions, to fully assess the potential benefits of these culture modifications, future experiments will explore the effects of the selected conditions (constant_30°C and Hs_37°C) on baculoviruses expressing a variety of recombinant proteins. This includes proteins regulated by early, late, or very late promoters, and under different MOI. Additionally, these studies may evaluate the quality of the produced proteins, including glycosylation profiles and secretion efficiency, to evaluate and determine the broader applicability of this approach.

We have updated the results section about the fluorescence analysis and the discussion section to emphasize these points and ensure their relevance and limitations are clear.

Comment 8 of 9 - Such observations have been made before and are the basis of a number of modified Sf9 cell lines stably expressing genes that aid in the expression of some proteins. I can't see this being referenced in the discussion. I think based on the results presented, the discussion and conclusion need to be modified to be more specific in the claims about the benefits of coexpressing/inducing heatshock proteins.

Thank you for your comment. We have revised the discussion section to address this point and provide a more specific and contextualized interpretation of our findings. This work represents a first approach to studying the impact that endogenous heat shock proteins and their overexpression, particularly HSP70 family proteins, may have in the baculovirus replication cycle and its productivity as a system for recombinant virus or proteins production.

While previous studies have linked these proteins to activities mainly in the early phase and replication of BV (Lyupina et al., 2010, 2011, 2014; Tung et al., 2016), others that cause recombinant Hsp70 co-expression have primarily focused on the chaperone effect of HSPs in enhancing recombinant protein folding and stability. Our findings suggest that improving hsp70s expression may have a broader effect, one positive effect in the baculovirus replication cycle itself (as observed through increased titers, BV-genomes, at least at 24 hpi, and accelerated production kinetics for the fluorescent reporters), while also, as a second effect, contributing to recombinant protein quality by its molecular chaperone activity, although this last was not probed in our experiments.

In the same direction, one key distinction of this proposal is the timing of HSP70 expression. It has been noted that chemical inhibition of the hsp70 induction or activity late in the baculovirus cycle, does not have the same effect as if this inhibition is done early in the infection (Lyupina et al., 2014). In a similar way, we expect that overexpression of these genes may have a bigger impact if it takes place before infection. Strategies that rely on very late promoters for HSP co-expression may have less of this effect and results more centered on the chaperone activity in the recombinant protein. In conclusion, we expect this earlier expression may provide benefits to both the viral cycle and the cellular environment supporting recombinant protein production.

There are two other aspects that may count for a difference, the first one is that inducing a stress response in the cell may implicate more genes and alterations than solely hsp70 genes. That could have positive effects in the cell, but sustained stress can have negative effects in productivity, probably as seen in the Hs_constant_30ºC, where at 72 hpi has the lowest values for PFU and reporters' fluorescence. In comparison to cell lines with chaperones, these are usually centered in improving the processing of the recombinant protein and are based in heterologous hsp70 or other chaperone genes.

Another aspect that we believe is a very important difference is that we are centering the study on endogenous HSP70 genes and trying to differentiate the activity of at least two different inducible members of this family. There are studies in mice where they have detailed a different phenotype for the knockout of each hsp70 gene, even when the level of similitude is high among these genes (Daugaard et al., 2007).

This brings to the table the possibility that each HSP70 gene in S. frugiperda may have a different role for the cell and a different effect in cases of its alteration. The same may apply to the Baculovirus and its cycle as it has been shown that these proteins are important for early events in its cycle. In this same line, we made an effort to clarify the identity and sequence of two inducible HSP70 genes (one of them with various very similar genes apparently in the same locus). This way we were able to monitor for differences in their expression in response to the infection or the combination of infection and previous heat shock. This led us to discover a higher expression by the group of hsp70Ab-group (in group), compared to hsp68 (approx. 6 times to 1), but it is in hsp68 where we found the highest and earliest changes in its expression 6 hpi.

As perspectives, thanks to the gene sequences obtained in this article, we aim to isolate and individually study the effects of the two different HSP70 genes on the baculovirus cycle and BEVS productivity. Given the sequence s

---

## [Decision Letter · Decision Letter 1]

27 Feb 2025

Molecular Characterization of the Effects of Heat Shock on the Infection Cycle Progression and Productivity of the Baculovirus Expression Vector System

PONE-D-24-50172R1

Dear Dr. Palomares,

We’re pleased to inform you that your manuscript has been judged scientifically suitable for publication and will be formally accepted for publication once it meets all outstanding technical requirements.

Kind regards,

Jian Xu, Ph.D.

Academic Editor

PLOS ONE

Additional Editor Comments (optional):

Reviewers' comments:

Reviewer's Responses to Questions

**Comments to the Author**

1. If the authors have adequately addressed your comments raised in a previous round of review and you feel that this manuscript is now acceptable for publication, you may indicate that here to bypass the “Comments to the Author” section, enter your conflict of interest statement in the “Confidential to Editor” section, and submit your "Accept" recommendation.

Reviewer #1: All comments have been addressed

2. Is the manuscript technically sound, and do the data support the conclusions?

Reviewer #1: Yes

3. Has the statistical analysis been performed appropriately and rigorously? 

Reviewer #1: Yes

4. Have the authors made all data underlying the findings in their manuscript fully available?

Reviewer #1: Yes

5. Is the manuscript presented in an intelligible fashion and written in standard English?

Reviewer #1: Yes

6. Review Comments to the Author

Reviewer #1: (No Response)

7. PLOS authors have the option to publish the peer review history of their article (what does this mean? ). If published, this will include your full peer review and any attached files.

**Do you want your identity to be public for this peer review?** For information about this choice, including consent withdrawal, please see our Privacy Policy .

Reviewer #1: No

---

## [Editor Report · Acceptance letter]

PONE-D-24-50172R1

PLOS ONE

Dear Dr. Palomares,

I'm pleased to inform you that your manuscript has been deemed suitable for publication in PLOS ONE. Congratulations! Your manuscript is now being handed over to our production team.

Kind regards,

on behalf of

Dr. Jian Xu

Academic Editor

PLOS ONE